



# Seasonal variability of radiation tide in Gulf of Riga

Vilnis Frishfelds[1], Juris Sennikovs[1], Uldis Bethers[1], and Andrejs Timuhins[1]

[1]Faculty of Physics, Mathematics and Optometry, University of Latvia, Riga, Latvia

**Correspondence:** Vilnis Frishfelds (frishfelds@latnet.lv)

**Abstract.** Diurnal oscillations of water level in Gulf of Riga are considered. It was found that there is distinct daily pattern of diurnal oscillations in certain seasons. The role of sea breeze, gravitational tides and atmospheric pressure gradient are analysed. The interference of the first two effects provide the dominant role in diurnal oscillations. The effect of gravitational tides is described both with sole tidal forcing and also in real case with atmospheric forcing and stratification. The yearly

variation of the declination of the Sun and stratification leads to seasonal intensification of gravitational tides in Gulf of Riga. Correlation between gravitational tide of the Sun with its radiation caused wind effects appears to be main driver of oscillations in Gulf of Riga. Daily variation of wind is primary source of $S_1$ tidal component with a water level maximum at 18:00 UTC in Gulf of Riga. Effect of solar radiation influences also $K_1$ and $P_1$ tidal components which are examined, too.

## 1 Introduction

A distinct feature in Gulf of Riga is diurnal oscillations of water level despite tidal influence is negligible in Baltic sea. These oscillations are especially expressed in a relatively calm and sunny spring days. The amplitude of oscillations can reach 10 cm of water level in May, see Fig. 1, that is much higher than typical amplitude of gravitational tides up to 3 cm (Medvedev et al., 2013). If we look on water level oscillations for longer period, then the resulting diurnal amplitude fades away as phase and modulation of these oscillations varies. Our aim here is to examine seasonal variations of amplitude and phase of diurnal

oscillations and find the dominating reason behind these variations.

    Moreover, observations show a distinct daily pattern in spring and summer when water level culminates in late afternoon, see Fig. 1. Therefore, there should be notable $S_1$ tidal component with oscillation period of exactly 24 h (Williams et al., 2018). Nature of $S_1$ tide is more related with diurnal changes in atmosphere caused by radiation rather than gravitational effect of the Moon and the Sun (Ray and Egbert, 2004). Therefore, $S_1$ is usually called as radiation or atmospheric tide.

Moreover, mechanism of radiation tide should be studied more in detail as water level oscillations may be driven by sea and land breeze or atmospheric pressure oscillations, which is the case for global ocean (Ray and Egbert, 2004). Radiation caused thermal expansion of surface water can yield only amplitude less than a millimeter in Gulf of Riga. Earth nutation is another factor influencing $S_1$ tide but the effect is rather weak comparing with atmospheric effects, see Schindelegger et al. (2016). Topographically trapped internal waves can be another factor influencing diurnal oscillations as in strait of Otranto (Ursella

et al., 2014) or Southern California Bight (Beckenbach and Terrill, 2008) but they could be important only equatorward of 30° latitude.





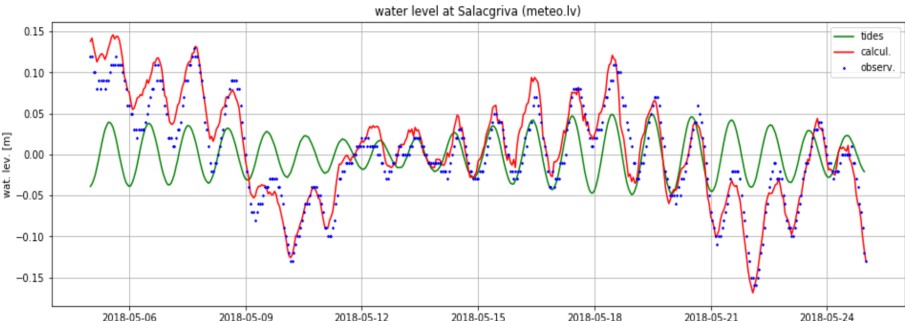

**Figure 1.** Water level oscillations in Salacgriva station in Gulf of Riga. Blue dots are observations; red curve – numerical reanalysis; green curve – simulated water level with only tidal forcing.

Diurnal oscillations of Gulf of Riga brings hundreds of km$^3$ through Irbe strait. This creates pulsating currents in the strait (Lilover et al., 1998). These pulsations do not bring notable water exchange between Baltic proper and Gulf of Riga as typical length traveled by water parcel result in $\approx 1$ km much shorter than the length of Irbe strait $\approx 45$ km. Several hours are required to transfer amount of water between Baltic proper and Gulf of Riga. Therefore, there is typical phase lag of oscillations in Gulf of Riga around 6-7 hours (Lilover et al., 1998) as compared to Baltic proper.

Hourly water level measurements in Gulf of Riga starts from 1960 in stations of Latvia. Sixty years of observations are sufficient for analysis as it is 3 times longer as lunar precession cycle of 18.6 years. Keruss and Sennikovs (1999) showed that the water level spectrum in Gulf of Riga really contains primarily diurnal tidal components with distinct peak of $S_1$ tidal component, see also Medvedev et al. (2013), whereas Baltic proper has relatively weak $S_1$ tidal component. Similar effects hold in Gulf of Finland and Curonian lagoon. Rabinovich and Medvedev (2015) associated $S_1$ component in Otkritoje of Curonian lagoon with sea breeze effect. Is it the case also for Gulf of Riga and Gulf of Finland? Figure 1 suggests that water level oscillations, i.e. phase and amplitude, correlates excellently with gravitational tides. Numerical reanalysis by HIROMB BOOS Model (HBM) at University of Latvia catches the oscillations very well, thus we can investigate the phenomenon by parameter studies. HBM has been used by several meteorological organisations around the Baltic sea and Copernicus Marine Monitoring Service (CMEMS) for operational analysis.

The period of diurnal tidal components $K_1$ (23.93 h) and $P_1$ (24.07 h) are nearly equal with $S_1$ tidal component. Therefore, there should be distinct interference between the solar radiation and gravitational tides, see Rabinovich and Medvedev (2015). Components $K_1$ and $O_1$ (25.82 h) are related to period of declination of the Moon. Components $K_1$ and $P_1$ are related to period of declination of the Sun. The influence of the Moon is higher, thus $K_1$ and $O_1$ are stronger than $P_1$. $K_1$ has usually similar amplitude as $O_1$ but it may depend on preferred oscillation period at given location. Position of the Moon may vary at solar noon when radiation is at peak. On the other hand, position of the Sun is principal characteristic of the solar noon. Therefore, we can expect that there can be correlation between gravitational effect of the Sun and radiation caused effects as sea breeze or atmospheric pressure variation.





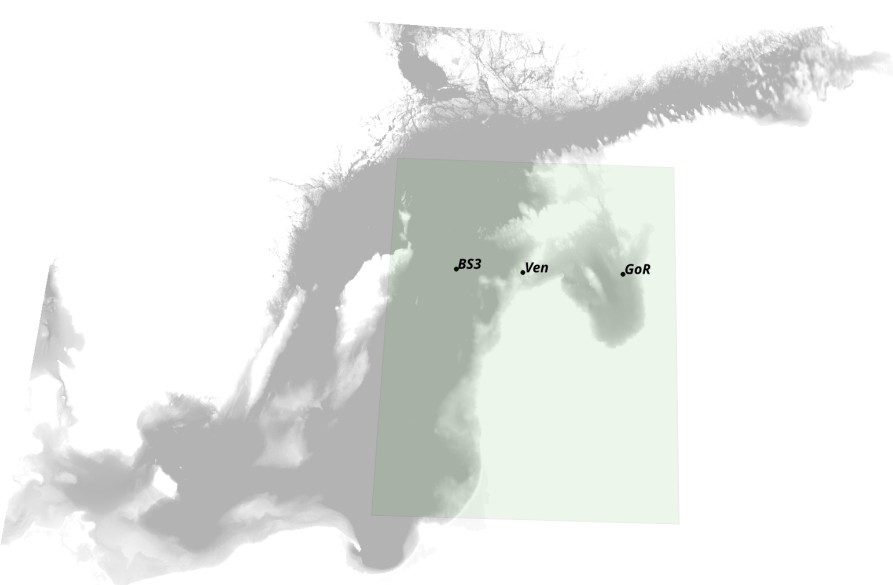

**Figure 2.** Simulation area of 3D oceanographic model HBM. Wide area has a resolution – 2 nmi (nautical miles), shaded area – 1 nmi. Locations of ERA-Interim analysis: $GoR$, $Ven$ and $BS3$ are marked by black points.

The main task of the study is to obtain and analyse the daily pattern of water level oscillations for every month in Gulf of Riga. Resonance of water level oscillations in the Baltic Sea will be studied in next section. Afterwards, spectral properties of water level observations will be summarised. Next, the role of gravitational forcing by the Sun and the Moon will be discussed. And finally, we will put radiation effect of the Sun together with gravitational forcing.

## 2    Oscillation Baltic proper - Gulf of Riga

The observations in Fig. 1 show that Gulf of Riga in connection with Baltic proper prefers to oscillate with nearly constant period of about 1 day but semi-diurnal components are effectively filtered out. Thus, there should be resonant nature of these oscillations that could be figured out by a numerical model, see Webb (2014). Otsmann et al. (1999) associates diurnal oscillations in Gulf of Riga and Väinameri sea with free Helmholtz oscillations of connected water bodies of Baltic proper, Gulf of Riga and Väinameri sea neglecting frictional and Coriolis forces. Assuming that the main connection of the Baltic sea is by
Irbe strait the cyclic frequency is given by:

$$\omega_{Gulf} = \sqrt{\frac{gA_{Irbe}}{L_{Irbe}A_{Gulf}}}, \tag{1}$$

where $A_{Irbe}$ and $L_{Irbe}$ are area of cross-section and length of Irbe strait, respectively, $A_{Gulf}$ is area of the Gulf of Riga and $g$ is gravitational acceleration. This gives about 24 hour period. Quarter-wavelength resonance theory can sometimes fail to give correct resonance pattern, see Cui et al. (2019). Instead, let us obtain the resonance pattern numerically by small harmonic


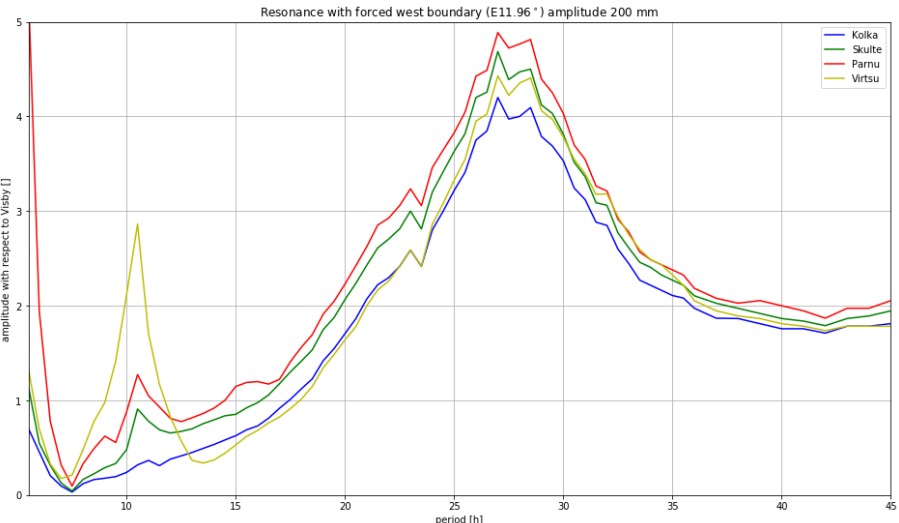

**Figure 3.** Amplitude of free oscillations vs. period in stations around Gulf of Riga with respect to Visby station.

water level perturbations far from Gulf of Riga and see how the water level in Gulf of Riga responds. There are numerous possibilities how to do insert the perturbations. Let us induce small water level oscillations in a distant location and calculate the water level variations with 3-dimensional HBM numerical model. Any other forcing is disregarded. Water temperature and salinity are set to be constant (density is 1007 kg/m$^3$) with no stratification. The model setup includes Baltic proper, Gulf of Riga and Gulf of Finland with 1 nautical mile resolution of Gulf of Riga, see Fig. 2. The resolution of Irbe and Suur straits

is very important factor as they have strong influence on the frequency of seiches, see Jönsson et al. (2008). The water level perturbation is placed at far west boundary of longitude of E 11.96°.

Baltic proper has its own response to perturbations on western boundary with one peak at 16 hours. We are interested to resonance pattern in Gulf of Riga relative to perturbations in Baltic proper in order to see oscillation associated with exchange of water through Irbe strait. Therefore, let us draw resonance pattern in Gulf of Riga with respect to Visby station (Gotland)

representing Baltic proper. Gotland is located close to amphidromic point of Baltic sea, thus it has rather weak resonance amplitudes both in semi-diurnal and diurnal range.

As can be seen in Fig. 3, Gulf of Riga has distinct peak with eigen period of 26-27 hours. That is a bit higher than analytic estimate 24 hours by Otsmann et al. (1999). But adding a Coriolis force generally increases period for longer seiche modes, see Leppäranta and Myrberg (2009). The amplitude and phase is almost equal in all stations around Gulf of Riga. This means that

Gulf of Riga oscillates as a single body. The phase lag with respect to Baltic proper is around 6 hours for diurnal oscillations that agrees with estimates by Lilover et al. (1998), see Fig. 4 where phase difference with respect to Ventspils station is shown. The highest amplitude is in Pärnu and the lowest in Kolka. As Virtsu station is close to Väinameri sea, then it has a semidiurnal peak associated with Väinameri oscillations. Väinameri sea have both semidiurnal and diurnal tidal components equally pronounced, see Otsmann et al. (1999). Seiches at Gulf of Finland have a period of 28 hours consistent with other numerical estimations





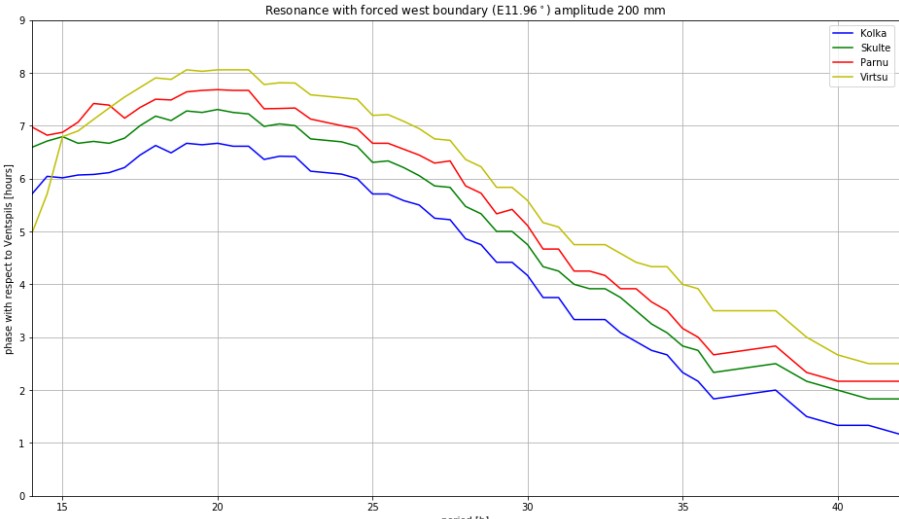

**Figure 4.** Phase difference of oscillations in Gulf of Riga with respect to Ventspils station (just outside Gulf of Riga).

by Wübber and Krauss (1979); Jönsson et al. (2008). But there are also strong semi-diurnal seiches, especially at Kronstadt and St.Petersburg. The presence of various oscillation modes with different periods and turbulent friction forces leads to strong damping of oscillations in Baltic sea, see Jönsson et al. (2008). Therefore, seiches with distinct amplitude are created only if there is a regular forcing such as gravitational tides, regular cyclonic activity, land and see breezes, Jönsson et al. (2008); Metzner et al. (2000). Regular forcing with period close to resonant one can be the primary source of oscillations in Gulf of

Riga, see Medvedev et al. (2016). The role of periodic forcing will be studied in next sections.

## 3   Water level observations

There are number of studies of spectral properties of water level time series in Baltic Sea. Let us summarize the Fourier analysis of water level on the basis of current set of observation data in eastern part of Baltic sea in order to list the principal tidal components, see Table 1. The results are consistent with earlier studies that diurnal components are particularly outspoken

in Gulf of Riga while Baltic proper and Väinameri Sea have more pronounced semidiurnal components. Strongest tides are in eastern part of Gulf of Finland, especially at Kronstdadt station, where semidiurnal components are significant, too. The water level observations of stations in Latvia are taken from LEGMC (Latvian Environmental and Geographical Monitoring Centre) starting from 1960 to 2019.The water level data from other stations are taken from Marine Copernicus service with data starting from 1980 or later. Observation data shows that $S_1$ component has comparable amplitude with diurnal components $K_1, O_1, P_1$.

Lunar component $O_1$ slightly exceeds luni-solar component $K_1$ in Gulf of Riga. It means that Gulf of Riga prefers to oscillate at slightly higher period than 24 hours as figured out in previous section. The long term Fourier analysis alone cannot provide the origin of the $S_1$ peak. Moreover, direct Fourier analysis of long time series gives only yearly averaged tidal amplitudes





**Table 1.** Amplitude (cm) of main tidal components in selected stations in eastern part of Baltic sea according to observations 1961-2019

|  | $M_2$=12.42 h | $K_1$=23.93 h | $S_1$=24 h | $P_1$=24.07 h | $O_1$=25.82 h | total |
|---|---|---|---|---|---|---|
| Kolka | 0.4 | 1.1 | 0.6 | 0.3 | 1.4 | 1.9 |
| Daugavgriva | 0.6 | 1.6 | 1.1 | 0.6 | 1.7 | 2.7 |
| Skulte | 0.4 | 1.2 | 0.8 | 0.5 | 1.4 | 2.2 |
| Salacgriva | 0.4 | 1.2 | 0.8 | 0.2 | 1.2 | 1.9 |
| Liepaja | 0.2 | 0.6 | 0.5 | 0.3 | 0.6 | 1.1 |
| Ventspils | 0.6 | 0.7 | 0.3 | 0.3 | 0.6 | 1.2 |
| Triigi | 1.4 | 0.7 | 0.5 | 0.3 | 1.2 | 2.2 |
| Visby | 0.5 | 0.4 | 0.3 | 0.1 | 0.3 | 0.9 |
| Kronstadt | 1.9 | 2.8 | 1.1 | 1.2 | 2.6 | 5.0 |

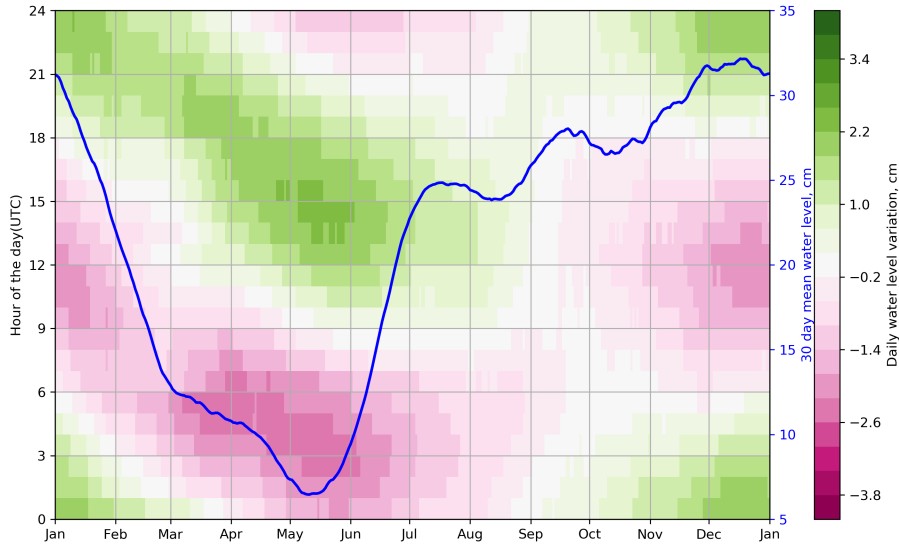

**Figure 5.** Daily variations of observations (1961-2019) at Skulte station. Average value of water level at respective time of the day (colour field). 30 days moving average water level (blue solid line).

without providing details about temporal variability, e.g., average amplitudes for each month. There are several sources of seasonal variability: stratification, ice build-up, sea and land breezes, climatology of wind, river inflow, distance to the Sun,
etc. Let us examine the daily pattern of oscillation in monthly scale. We can expect that due to solar radiation, highest daily temperature variations should be in June and lowest in December. However, Fig. 5 suggests that highest daily variations occur in late spring and the lowest in early autumn. That could not be described just by solar radiation. Thus, we need to account the gravitational tides as will be done in the next section. The time of maximum water level in Gulf of Riga is late afternoon



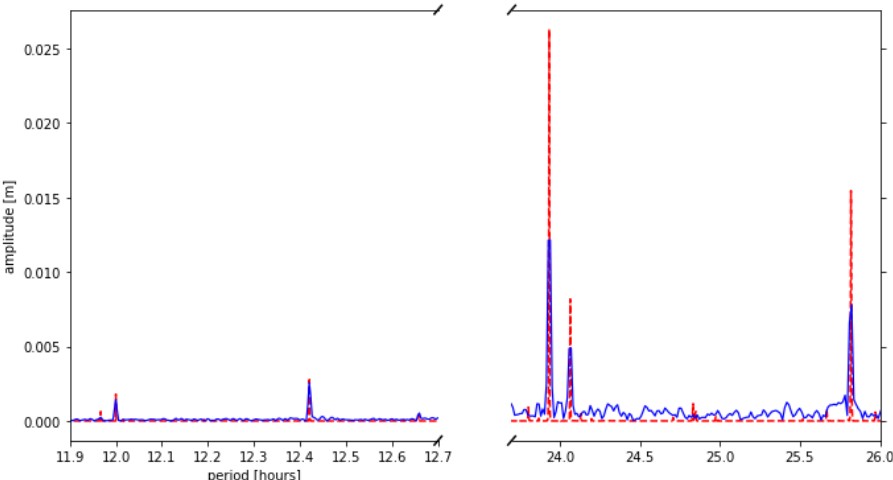

**Figure 6.** Simulated spectral intensity of water level in Skulte station. Red dashed curve – only gravitational forcing 1990-2020; blue – difference in real model with gravitational tides included and excluded (2014-2020).

throughout spring and summer but it shifts to midnight in autumn and early winter. Higher average water level in autumn and
winter are caused by presence of more active Atlantic cyclones during this period with predominant south-west winds. Low average water level in May is typical phenomenon in Baltic Sea. The characteristic seasonal variability of water level are nearly the same for all station around Gulf of Riga with highest amplitude in Pärnu and lowest in Kolka, see Fig. 3.

## 4   Gravitational tides

As noted in the previous section it is not enough to consider solar radiation alone to describe the seasonal variability of daily
changes of the water level. Another periodic forcing is provided by gravitational tides. First, let us consider that tides have purely gravitational origin due to the Sun and the Moon. Basing on ephemerides and ocean circulation model we can estimate amplitudes of tidal components. This is calculated by ocean circulation model HBM, where tidal stress is implemented through unresolved bottom shear, see Canuto et al. (2010). Calculations are performed in non-stratified conditions with constant homogeneous density. Purely gravitational tides do not lead to any notable $S_1$ component, see Fig. 6. Ray and Egbert
(2004) estimated that gravitational $S_1$ should be about 200 times smaller than $K_1$. Lack of $S_1$ component in simulations is compensated by increased $K_1$ component as compared to observations, see Table 1.

In order to study real conditions, it is better to evaluate the gravitational tides with atmospheric forcing, real temperature and salinity distributions, and real boundary conditions accounted. It can be evaluated by taking the difference between real case with tidal potential and without. This is done for the period 2014.07.01-2020.01.16. The two simulations may eventually yield
different distributions of temperature and salinity as tidal contribution enhances mixing of the water. Therefore, distribution of temperature and salinity in non-tidal simulation is fitted to tidal one after every few months.


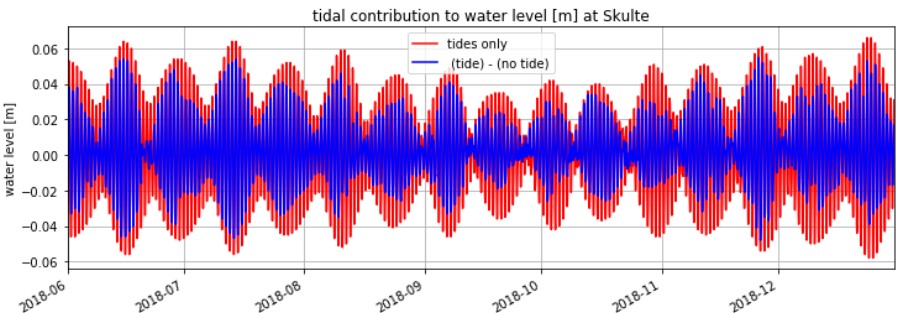

**Figure 7.** Tidal effect on water level of Skulte station. Red - tidal forcing only. Blue – difference of water level in cases of included and excluded tidal potential with real atmospheric forcing, temperature and salinity distribution.

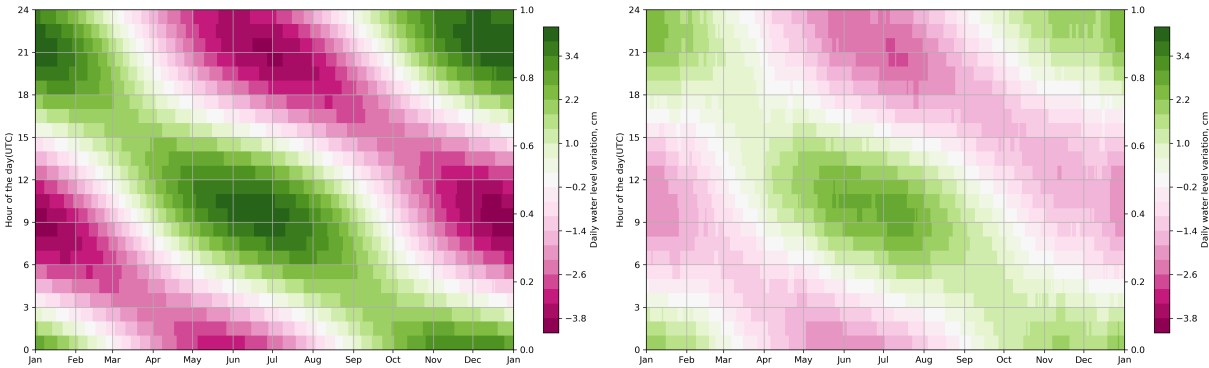

**Figure 8.** Daily variations of gravitational tides at Skulte station. Hour (UTC) of the maximal and minimal water level of the day (dashed lines) and analytic approximation (dash dotted line) vs. month. Average value of water level at respective time of the day (solid lines) and analytic approximation (dotted link). Average monthly water level (green solid line). Left: tidal forcing only (1990-2020), right: difference of water level in real case of included and excluded tidal potential (2014-2020).

As can be seen from Fig. 7, real atmospheric forcing tends to minimize the effect of gravitational tides but the pattern of oscillations remains nearly equivalent. Change of stratification is not the dominant characteristic as Irbe strait is shallow with maximal depth of 27 m. Such components as smaller lunar elliptic diurnal component $M_1$ (24.84 h) can be clearly seen in

Fig. 6, but noise both in observations or simulations with real forcing makes it hidden. Amplitude of semi-diurnal components does not depend much on whether purely gravitational forcing or real forcing is applied. The amplitude for diurnal components is almost twice lower in the second case.

Now, let us look on daily pattern of gravitational tides during every month. Presence of the Sun and the Moon makes water level highest at one time of the day and lowest at another time of the day. The effect of gravitational tides is strongest

in June-July and December-January Fig. 8, i.e., at winter or summer solstice. Lowest daily variation occurs at March and September, i.e., at spring or autumn equinox. The average daily tidal amplitude in June and December can reach 4 centimeters


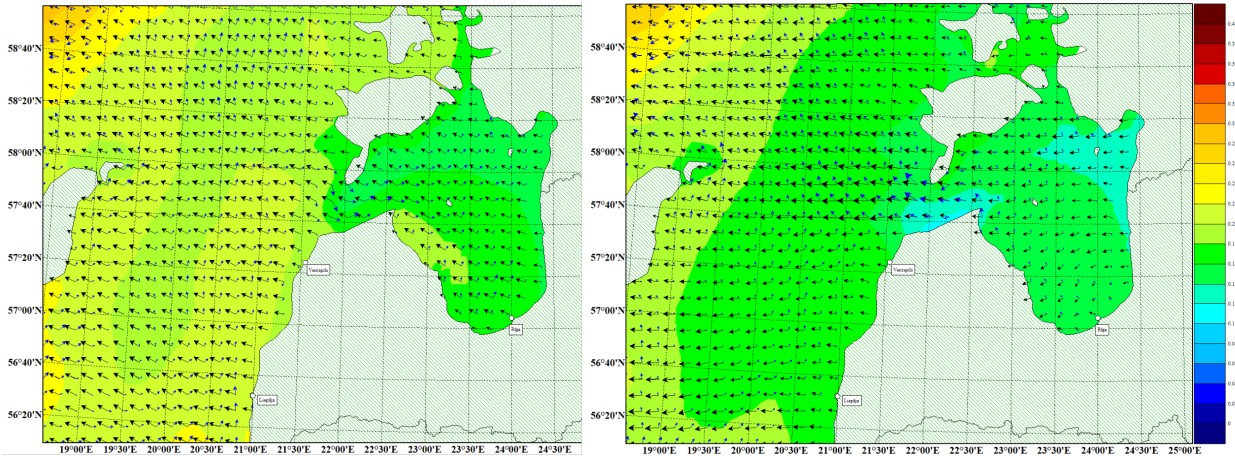

**Figure 9.** Wind and surface current at 2017.09.26 03:00 UTC (left) and 2017.09.26 15:00 UTC (right). Black arrows – 10 m wind, blue arrows – surface current, fill colour – water level.

with tidal forcing only and 3 centimeters in real case. Amplitude is a bit higher in December than in June with only tidal forcing because the Sun is slightly closer to Earth in December-January. However, tidal amplitudes are smaller in winter (well-mixed conditions) than during summer (stratified conditions) and this difference is up to 10 % (Müller, 2012). Figure 8 with

real conditions confirms that conclusion. The time of maximum water level is at solar noon in June and at solar midnight in December. If we look at particular day of summer or winter solstice each year then the moment of maximum water level is always ±2 hours from either solar noon or solar midnight, respectively. Thus, there is deterministic gravitational influence of the Sun despite the Moon has stronger effect on tides. In April-May the maximum water level is achieved few hours after the solar noon and vice versa in August-September. Water level influenced by gravity of the Sun can be represented by the form

$$S(t) \approx K \cos((\omega_d + \omega_y) * (t - t_0)) + P \cos((\omega_d - \omega_y) * (t - t_0)),\qquad(2)$$

where $t_0$ is time moment of water level maximum at noon of the day of summer solstice with gravity forcing of the Sun only; $K$ and $P$ are constants characterising $K_1$ and $P_1$ components, respectively. Data in Fig. 8 left yields $K$ as 3 cm and $P$ - as 1 cm and Fig. 8 right yields $K \approx 1.7$ and $P \approx 0.6$ cm, where $t_0$ corresponds to 10:30 UTC time in June 21. Now, we must check how the phase of gravitational forcing matches with effect of solar radiation.

**5  Correlation of gravitational tides with atmospheric forcing**

Daily oscillations of water level in Gulf of Riga are best expressed in case of long, calm and sunny periods without strong cyclones. Then, weather pattern is typically governed by light easterly wind. Low magnitude of winds suggests that sea breeze winds can contribute a notable part to the daily averaged wind. As example let us look on typical anticyclonic situation in Fig. 9. At 03:00 UTC time, there is relatively weak easterly wind that does not prevent the sea surface current to flow in





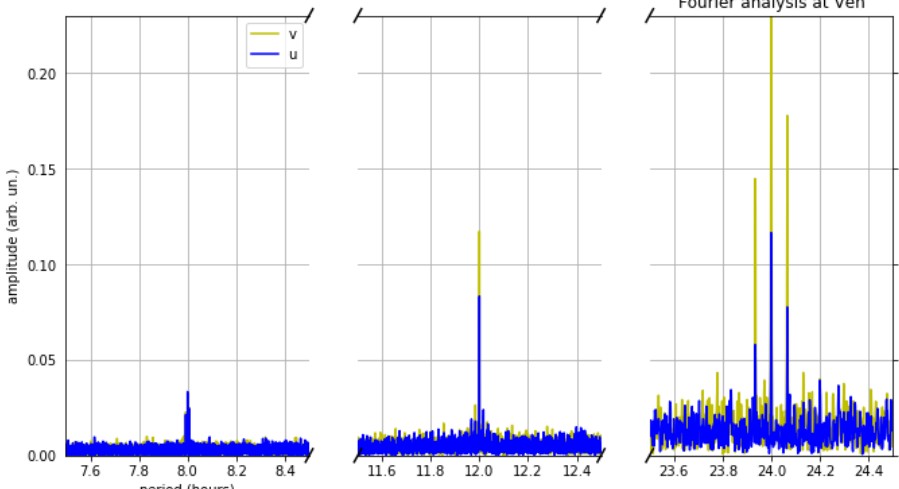

**Figure 10.** Fourier spectra of zonal (blue) and meridial (yellow) winds close to entrance of Irbe strait at N 57.75°, E 21.25° (point $Ven$ in Fig. 2).

opposite direction. However, easterly winds are stronger and surface current in Irbe strait is oriented along the wind at 15:00 UTC. In order to estimate long time sea breeze effect, we could use the observation data. However, long time continuous data with less than 3 hour interval are not available for marine locations. Therefore, let us use a reanalysis product, e.g., ECMWF ERA-Interim (Dee et al., 2011) for period 1979-2017. Fourier analysis of wind speed at location close to Irbe strait shows that diurnal components are present together with over modes at 12 and 8 hour periods, see Fig. 10. There are also oscillation

modes at 23.93 h (the same as $K_1$ tidal component) and 24.07 h (the same as $P_1$ tidal component) hour periods. Their presence suggests that daily variation of wind is much stronger in summer than in winter. Thus, radiation effects are important not only for $S_1$ but also $K_1$ and $P_1$ tidal components, see Medvedev et al. (2016). Now, let us check the pressure variations at various times of the day, see Fig. 11. The average surface pressure is nearly 1 mbar higher in Gulf Riga than in Baltic proper in late autumn and winter due to lower Atlantic influence. Figure 11 confirms that at the time of the daily maximum air temperature,

surface pressure is falling at its fastest rate (Cook, 2012). The reanalysis data also confirms that sea breeze winds peak at the maximum temperature with a small lag, less than one hour (Cook, 2012). Daily change of pressure is stronger in Baltic proper than in Gulf of Riga, i.e., the difference between these points is mainly due to pressure change in Baltic proper. Notable daily pressure variations occur between March and October. The amplitude of mean sea level pressure difference is only 30 Pa, which can result in daily amplitudes of water level of Gulf of Riga with less than 1 cm by barotropic pressure gradient. The

anti phase of pressure difference, see Fig. 11, is roughly one to two hours before phase of water level in Gulf of Riga. Such a small lag time is not enough to fill Gulf of Riga. Therefore, daily water level oscillations can be caused primarily by sea and land breezes.





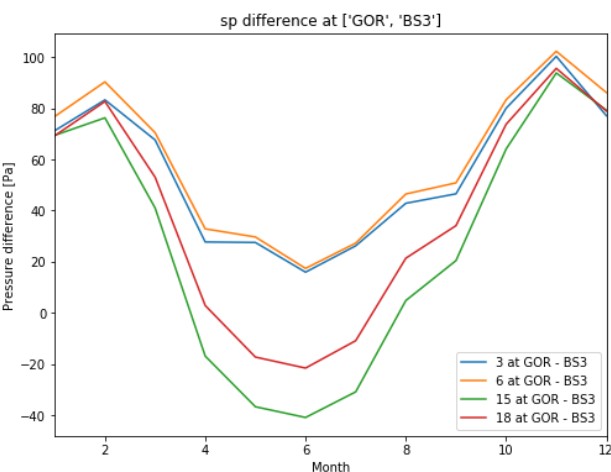

**Figure 11.** Monthly mean sea level pressure difference [Pa] between locations (see Fig. 2) $GoR$ – N 57.75°, E 23.75° and $BS3$ – N 57.75°, E 20.00° at various times per day [UTC] according to ERA-Interim reanalysis.

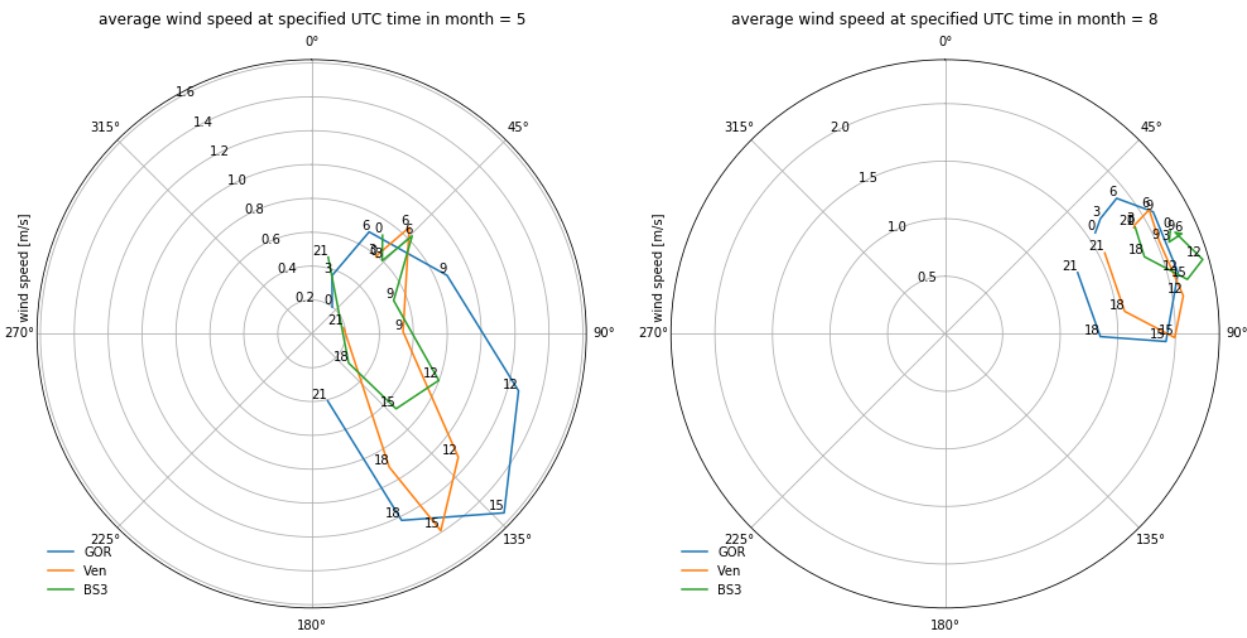

**Figure 12.** 10 m wind hodograph (UTC time) from ERA-Interim reanalysis at latitude N 57.75° and longitude at locations (see Fig. 2): $GoR$ - E 23.75°, $Ven$ - E 21.25°, $BS3$ - E 20.00° in specified month. Left – May, right – August.

Magnitude of sea breeze wind oscillations can be estimated by $\sqrt{\Delta T}/4$ (Miller et al., 2003). The land – sea temperature difference $\Delta T$ can reach about 10 degrees in May, which result in about 1 m/s amplitude of sea and land breeze, which agrees with our findings, see wind hodograph as in Moisseeva and Steyn (2014) in Fig. 12. Actually, the difference in wind speed





is not as large to call the effect as "breeze" but it helps to interpret the results. Coriolis force in northern hemisphere leads to clockwise daily variation of wind direction (Miao et al., 2009). The amplitude of breeze is stronger in Gulf of Riga than in central Baltic proper as the breezes are stronger at the interface between land and sea. It is interesting that amplitude of daily sea and land breezes is weakly dependent on mean sea level pressure. Approximately the same amplitude of daily variation of wind occurs in cases of high pressure with typically light easterly winds as for low pressure situations with stronger westerly winds generated by Atlantic cyclones. The phase of daily variation of water level is consistent with phase of land and sea breezes accounting time lag between these. Easterly wind cedes quickly after 12:00-15:00 UTC, see Fig. 12. Thus, the time of maximum water level in Gulf of Riga neglecting tidal influence is at about 18:00 UTC.

Further, combining the effect of sea and land breezes with gravitational tides (Fig. 8) we can explain the seasonal variability of daily water level variations in Fig. 5. Atmospheric forcing is nearly in phase with gravity of the Sun in March-May that yields highest daily variation of water level. This occurs despite both the gravitational tides are stronger in June-July and the effect of daily wind variation is at peak. The reason of lower daily variations in late summer and especially September, is that sea and land breeze is almost in anti-phase with gravitational tides and they cancel each other. In December, there is almost no effect from solar radiation but there is strong influence of solar gravity when the Sun is close to antipodal position in midnight.

Let us approximate the variation of radiation tide by:

$$R(t) \approx \cos(\omega_d(t - t_0 - \Delta t))(A + B\cos(\omega_y(t - t_0))), \tag{3}$$

where $\omega_d$, $\omega_y$ are cyclic frequencies corresponding to 24 h and tropical year periods, respectively; $\Delta t$ is time interval between maximum water level created by solar radiation and gravity forcing of the Sun at day of summer solstice; $A$, $B$ are constants. The approximation assumes symmetric behaviour with respect to solstice. The product can be written as:

$$R(t) \approx A\cos(\omega_d(t - t_0 - \Delta t)) + \frac{B}{2}\cos((\omega_d + \omega_y)(t - t_0) - \omega_d\Delta t) + \frac{B}{2}\cos((\omega_d - \omega_y)(t - t_0) - \omega_d\Delta t). \tag{4}$$

Combining with gravitational approximation in Eq. (2) the total change of water level becomes

$$R(t) + S(t) \approx A\cos(\omega_d(t - t_0 - \Delta t)) + E\cos((\omega_d + \omega_y)(t - t_0) - \alpha_1) + F\cos((\omega_d - \omega_y)(t - t_0) - \alpha_2), \tag{5}$$

where $\alpha_1$ and $\alpha_2$ are phase constants and resulting amplitudes, i.e., tidal components $K_1$ and $P_1$ are

$$E = \sqrt{K^2 + 2BK\cos(\omega_d\Delta t) + B^2}, F = \sqrt{P^2 + 2BP\cos(\omega_d\Delta t) + B^2}. \tag{6}$$

$\Delta t$ is roughly about 7-8 hours in Gulf of Riga according to Fig. 8 and Fig. 5. Thus, $\cos(\omega_d\Delta t) \approx -0.4$. Therefore, radiation tide slightly changes the amplitudes of both components $K_1$ and $P_1$. According to observations $A \approx 1$ cm and $B \approx 0.8$ cm in Skulte station that are between gravitational amplitudes of $K \approx 1.7$ cm and $P \approx 0.6$ cm in Eq. (2). Thus, gravitational tides have still larger contribution in diurnal oscillations than see breeze. Nevertheless, phase difference between them is the main source of daily pattern of diurnal oscillations in Gulf of Riga from spring to autumn equinox. Results for station Skulte are summarised in Fig. 13. Despite the approximation by Eq. (5) is rather simple, it qualitatively well describes daily water level variation in every season.



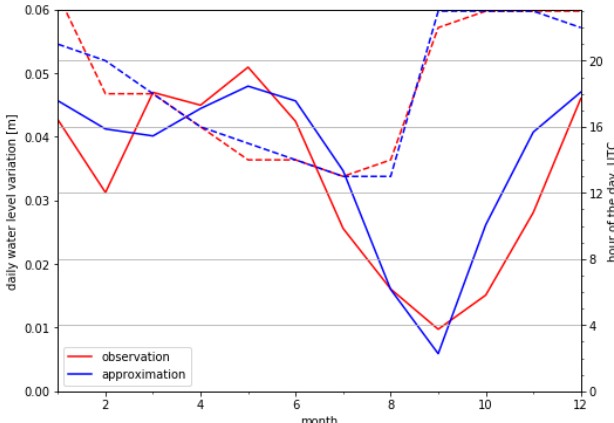

**Figure 13.** Water level daily variation (solid lines) each month at Skulte. Dashed lines denote the time of maximal water level. Red lines represent observation, blue - approximation by Eq. (5).

# 6 Conclusions

Gulf of Riga experiences distinct diurnal mode of seiches whose period is mainly determined by the hydrodynamic resistance in Irbe strait. Väinameri sea laying in the north has also semidiurnal oscillations.

Phase lag between Gulf of Riga and Baltic proper is typically about 6 hours for all locations in the Gulf of Riga. The phase is almost the same in all stations around Gulf of Riga with highest water level amplitude in Pärnu and lowest in Kolka along south eastern coast.

Observations suggest presence of distinct $S_1$ tidal mode in Gulf of Riga. Purely gravitational tides do not cause appearance of $S_1$ tidal mode despite strong daily pattern both in June and December. Atmospheric influence causes appearance of not only

$S_1$ component but influences $K_1$ and $P_1$ tidal modes.

Real atmospheric forcing tends to minimize the effect of gravitational tides. The influence of stratification is minimal for the Gulf of Riga with respect to gravitational tides as the shallow Irbe strait is usually well mixed.

There are distinct daily mean sea level pressure variations in Gulf of Riga and Baltic proper with gradient of fraction of millibar per 100 km. This is, however, too small to cause significant daily water level oscillations in Gulf of Riga. Daily mean

sea level pressure variations result from sea and land breeze effect.

There is a distinct sea and land breeze effect in Gulf of Riga with daily amplitude of around 1 m/s in spring-summer. It results that water level in Gulf Riga culminates in late afternoon when the sea breeze is ceasing. That is not the case for autumn and winter when the daily phase of water level is related to cyclones. The interpretation of daily wind variation as sea or land breeze is only approximate as actual wind directions may be arbitrary.

Average daily water level variations are strongest in April-May when the solar gravity is in phase with sea breeze effect. In contrast, average daily water level variations are lowest in September when gravitational tides are almost in anti-phase with effects of solar radiation.





*Author contributions.* All authors have contributed to the paper. The main responsibility of VF was running ocean simulations and gathering data. JS and UB are senior researchers in field of oceanography and provided valuable comments. JS and AT were responsible for Fourier

analysis, observation data and programming in Python.

*Competing interests.* The authors declare that they have no conflict of interest.

*Acknowledgements.* Authors are grateful to support by Latvian Fundamental and Applied Research Project lzp-2018/1-0162.





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
