# Peer review of "Seasonal variability of radiation tide in Gulf of Riga"

_Ocean Science, 2020_

## Referee Comment (RC1) · Anonymous Referee #1 · 24 Feb 2020

Causes for diurnal oscillations in the water level in the Gulf of Riga are investigated, and a prominent role of tides and sea breezes is postulated.

In its current form I have to advise that this paper is rejected by Ocean Science. The primary reasons for this is that it is very hard to follow the logic of the paper, to understand how the experiments were carried out and to understand what the figures show. Some examples are that at times it appears that the authors argue that atmospheric forcing makes tides (or perhaps diurnal oscillations in general) weaker, and at other times stronger. Tidal amplitudes are calculated from a model (at times), and it says that in the model tidal stress is implemented through unresolved bottom shear, see Canuto. I have no ideas what that means. Moreover, the usage of citations like in the aforementioned example where things are said to be this or that according to him/her

without giving a proper explanations is problematic throughout the paper. Lastly, several figs are said to show daily variations of sea level, and they are often negative. I would have expected that a variation to be a positive quantity (e.g. a standard deviation), I don't find the definition of variation in the paper. I also can't follow why the authors believe S1 to be more important than K1 and O1.

To a large degree the problems of the paper may be owing to linguistic shortcomings, and the paper requires substantial work to improve both readability and traceability.

---

## Referee Comment (RC2) · Anonymous Referee #2 · 25 Feb 2020

The paper discusses mechanisms behind the diurnal oscillations in the water level in the Gulf of Riga. This is an important topic in Baltic Sea research. The paper claims that the Helmholtz oscillations, which are normally discussed as the main reason of oscillations in the literature, can not completely explain the observed phenomenon and argue that the tidal components contribute significantly in the oscillations. The authors attempt to use numerical modelling instead of an analytical approach to study the origin of the oscillations.

However, the paper lacks logic in its structure, is written in the inconsistent matter, and is very difficult to read due to the language problems. The research is not scientifically robust, inconclusive and not well described. Therefore, the paper is not recommended for publication in the Ocean Science journal in its current shape.

[Figure]

As example of the illogical structure, the authors show the Figure 1 in the introduction as the reason for the presence of the tidal components in the water level variability. However, this figure is based on the modelling results described much later in the paper. Also, the graph shows the observations without long-term variability, but there is no information about how exactly the long-term components were estimated. The article needs to be significantly re-written following the standard logics of a scientific paper so that in the introduction only the previous research on the topic is discussed, then all the methods, techniques and modelling parameters are discussed, followed by the results presented in the results section. At the moment, almost all the parts of the paper are mixed.

The methods should be carefully described in the paper. For example, which modelling parameters were used? How were the long-term components removed? What is the shape of the perturbation introduced in the HBM model? Why was it introduced only at longitude = 11.96? How changing the parameters of the perturbation or their location affect the results?

---

## Author Comment (AC1) · 23 Mar 2020

> Causes for diurnal oscillations in the water level in the Gulf of Riga are investigated, and a prominent role of tides and sea breezes is postulated.

OK

> In its current form I have to advise that this paper is rejected by Ocean Science. The primary reasons for this is that it is very hard to follow the logic of the paper, to understand how the experiments were carried out and to understand what the figures show.

Text of the paper is reworked. The task of the paper is expressed more clearly in abstract, introduction and analysis. Resonance of the Gulf of Riga is moved to Appendix

as its is less substantial part for the main task of finding characteristics of daily pattern of oscillations of water level for every season. Then, there remain three pricipal sections: a) observations of daily water level oscillations, b) case with astronomical tides only, c) inclusion of sea and land breeze and analysis.

> Some examples are that at times it appears that the authors argue that atmospheric forcing makes tides (or perhaps diurnal oscillations in general) weaker, and at other times stronger.

The language is improved, now. Yes, it depends on the context. Astronomical tides are of course stronger, but presence of sea breeze is essential in daily phase of water level oscillations in spring and summer.

> Tidal amplitudes are calculated from a model (at times), and it says that in the model tidal stress is implemented through unresolved bottom shear, see Canuto. I have no ideas what that means.

Canuto divided tidal influence as 3 constituents: tidal drag in shallow seas, internal baroclinic tides and unresolved bottom shear. The last one is component of tidal field not aligned with the mean velocity and cannot be modeled as a tidal drag. It is principal constituent (according to tests) because of weak flow field with only tidal forcing. Description added in the text.

> Moreover, the usage of citations like in the aforementioned example where things are said to be this or that according to him/her without giving a proper explanations is problematic throughout the paper.

Problematic citations have been explained better.

> Lastly, several figs are said to show daily variations of sea level, and they are often negative. I would have expected that a variation to be a positive quantity (e.g. a standard deviation), I don't find the definition of variation in the paper.

Yes, "variation" was not a proper expression. Oscillation or deviation of water level was
intended. Corrected in text.

> I also can't follow why the authors believe S1 to be more important than K1 and O1.

Yes, O1 and K1 are stronger. Figure with spectral distribution of diurnal components from observations (1961-2019) is added (attached). But since we would like to find an hour of the day of particular month of unspecified year with either maximum or minimum water level, then lunar tides should be disregarded and S1 becomes essential component in spring-summer. The task and motivation of the paper is expressed better in abstract, introduction and discussion. Basically, the task is to get the observed daily pattern of water level oscillations at each month (Figure 5 in version 1), and test whether it can be explained only with astronomical tides and sea breeze; and find the proportions of their contribution (Figure 13 in version 1).

> To a large degree the problems of the paper may be owing to linguistic shortcomings, and the paper requires substantial work to improve both readability and traceability.

Yes. The structure of the text of paper is reworked, simplified and unnecessary phrases less related to the subject are removed.

[Figure]

**Fig. 1.**

---

## Author Comment (AC2) · 23 Mar 2020

> The paper discusses mechanisms behind the diurnal oscillations in the water level in the Gulf of Riga. This is an important topic in Baltic Sea research.

OK

> The paper claims that the Helmholtz oscillations, which are normally discussed as the main reason of oscillations in the literature, can not completely explain the observed phenomenon and argue that the tidal components contribute significantly in the oscillations.

Semi-closed state of Gulf of Riga just ensures that only diurnal components survive there. Tides (astronomical and radiation) must be primary forcing that can drive these

periodic oscillations with strong damping. Description improved in text.

> The authors attempt to use numerical modelling instead of an analytical approach to study the origin of the oscillations.

Yes, since it is hard to estimate analytically the damping effect in Irbe strait and particular geometry. Description improved in text.

> However, the paper lacks logic in its structure, is written in the inconsistent matter, and is very difficult to read due to the language problems. The research is not scientifically robust, inconclusive and not well described. Therefore, the paper is not recommended for publication in the Ocean Science journal in its current shape.

Yes, the structure of the paper is improved with better expressed task of the paper. Resonance of the Gulf of Riga is moved to Appendix as its is less essential part for the main task of finding seasonal characteristics of daily pattern of oscillations of water level. Then, there are three principal sections: a) observations of daily water level oscillations, b) case with astronomical tides only, c) inclusion of sea and land breeze and analysis.

> As example of the illogical structure, the authors show the Figure 1 in the introduction as the reason for the presence of the tidal components in the water level variability. However, this figure is based on the modelling results described much later in the paper.

Figure 1 was intended just as example. Of course, we cannot put 60 years of observations as readable figure. Therefore, spectral analysis of observations (1961-2019) will be included. Better description added in text.

> Also, the graph shows the observations without long-term variability, but there is no information about how exactly the long-term components were estimated.

The long term components were studied primarily by spectral analysis. Similar result with maximum contribution of S1 oscillations in May and minimum in September was

obtained by using Lomb-Scargle periodogram and short time Fourier analysis. But they would not yield any better information as simple hourly statistics for the main task (Figure 5 in version 1) and not included in the text. The description of observations is improved.

> The article needs to be significantly re-written following the standard logics of a scientific paper so that in the introduction only the previous research on the topic is discussed, then all the methods, techniques and modelling parameters are discussed, followed by the results presented in the results section. At the moment, almost all the parts of the paper are mixed. The methods should be carefully described in the paper. For example, which modelling parameters were used? How were the long-term components removed?

Yes, the language is improved and task of the paper is being expressed more clearly in the text. Better description of modelling parameters is being added.

> What is the shape of the perturbation introduced in the HBM model? Why was it introduced only at longitude = 11.96? How changing the parameters of the perturbation or their location affect the results?

Perturbation is harmonic with given period. Since there is no other forcing, after few days the system will converge to a solution where any point will oscillate with the period of perturbation but with specific amplitude and phase. Then, basing on the local water level amplitudes we can conclude whether specific part of the basin wants to oscillate with given frequency. Since, the relative amplitude is used with respect to Visby station, then it should not depend on location of perturbation. The same is true also for relative phase. We have used various resolutions and different setups (nested and non-nested) all giving the same picture. The description is improved in the text.

---

## Author Comment (AC3) · 26 Mar 2020

Correction of answer to interactive comment on "Seasonal variability of radiation tide in Gulf of Riga" by Vilnis Frishfelds et al. Anonymous Referee #1

> Tidal amplitudes are calculated from a model (at times), and it says that in the model tidal stress is implemented through unresolved bottom shear, see Canuto. I have no ideas what that means.

Correction as the first answer was incomplete. Yes, the tidal calculations by the model are used to test that observed peaks in water level spectrum can be described by astronomical tides with exception of S1. And also K1 and P1 components can be differ. Therefore, observed water level spectrum will be included in order to compare

with spectrum obtained by the model which includes only astronomical forcing. Tidal potential in the model is obtained basing on the actual ephemerides of celestial bodies (Sun and Moon). Ephemerides includes hourly data for right ascension, declination and distance between celestial body and centre of Earth. Gradient of tidal potential provides forcing in momentum equation of ocean circulation. But there are other effects of tides: enhanced mixing and additional drag at the bottom. Mixing is unimportant for homogeneous water, but bottom shear by Canuto at low flow velocities becomes an important factor. Description of numerical model with tidal forcing will be supplemented and corrected in text.